# Encoding extracellular modification of artificial cell membranes using engineered self-translocating proteins

Alexander Harjung ⓘ, Alessandro Fracassi ⓘ & Neal K. Devaraj ⓘ ✉

The development of artificial cells has led to fundamental insights into the functional processes of living cells while simultaneously paving the way for transformative applications in biotechnology and medicine. A common method of generating artificial cells is to encapsulate protein expression systems within lipid vesicles. However, to communicate with the external environment, protein translocation across lipid membranes must take place. In living cells, protein transport across membranes is achieved with the aid of complex translocase systems which are difficult to reconstitute into artificial cells. Thus, there is need for simple mechanisms by which proteins can be encoded and expressed inside synthetic compartments yet still be externally displayed. Here we present a genetically encodable membrane functionalization system based on mutants of pore-forming proteins. We modify the membrane translocating loop of α-hemolysin to translocate functional peptides up to 52 amino acids across lipid membranes. Full membrane translocation occurs in the absence of any translocase machinery and the translocated peptides are recognized by specific peptide-binding ligands on the opposing membrane side. Engineered hemolysins can be used for genetically programming artificial cells to display interacting peptide pairs, enabling their assembly into artificial tissue-like structures.

Living cells are compartmentalized by phospholipid membranes, which separate the contents of a cell from the extracellular environment and provide a barrier that is crucial for cell survival and identity. However, lipid barriers also limit interaction and communication of cells with the extracellular environment since cellular membranes are generally not permeable to most biological molecules[1]. To resolve this issue, cells have evolved many complex mechanisms that enable modification and functionalization of their membranes. This functionalization is often achieved through the insertion of transmembrane proteins which serve as a connection between the intracellular and extracellular environment and facilitate nutrient uptake, signaling, and tissue formation[2]. Membrane proteins, due to their amphipathic nature, rarely spontaneously insert into membranes. Instead, these proteins achieve proper topology and folding within the membrane with

the aid of highly evolved cellular translocase systems. An important example of such a mechanism is the Sec61-dependent protein translocation pathway in which several proteins form a translocation channel enabling protein insertion across a membrane[3].

Recently there has been increasing interest in developing artificial cells as models that mimic the structure and function of living cells. Perhaps the most common method for generating artificial cells is to encapsulate proteins within vesicles consisting of phospholipid membranes. Advances over several decades have enabled researchers to routinely form membrane-bound vesicles and even encapsulate the necessary biological components for cell free protein synthesis[4]. However, like in living cells, the lipid membranes of artificial cells constitute a barrier that prevents interaction of internally expressed proteins with other cells or the external environment. Given the

Department of Chemistry and Biochemistry, University of California, San Diego, CA, USA. ✉e-mail: ndevaraj@UCSD.edu

importance of membrane proteins in living cells, there exists a major need for simplified mechanisms by which proteins internally expressed within vesicles can embed themselves into membranes such that the protein translocates through the phospholipid-bilayer. If a significant part of the protein is exposed on the outer leaflet, interactions with the external environment of the artificial cell would then become possible.

While advances have been made in reconstituting the natural protein insertion machinery in protocells, this approach remains highly complex and has limited efficiency[5]. Alternatively, researchers have developed methods for functionalizing artificial membranes by doping the membrane with an affinity ligand. This ligand can then serve as a handle by which the membrane can be chemically functionalized using proteins bearing appropriate ligand binding groups[6]. However, in natural cells, membrane modification is self-encoded as part of the genome of the cell. In addition, it has been shown that using in vitro expression systems in the presence of lipid bilayers can lead to spontaneous insertion of some membrane proteins into lipid bilayers[7,8]. However, it is not clear if this in situ membrane protein insertion process leads to full translocation of the lipid bilayer or if such proteins can mediate extracellular interactions.

One strategy that could enable external membrane modification of individual artificial cells using genetically encoded proteins is to take advantage of soluble proteins that are able to self-translocate across lipid membranes. There are some rare examples of proteins, such as pore forming toxins (PFTs) which can self-insert into biological membranes, independent of any insertion machinery[9–14]. A well-studied example of a PFT is the bacterial toxin α-hemolysin (αHL). Pore formation occurs when αHL binds to the membrane, first as a soluble monomer, and then subsequently forming a heptameric complex, which spontaneously translocates across the lipid membrane as a barrel-like structure[15]. αHL has been extensively studied due to its biological role in infection and its utility in nanopore sequencing. Due to its self-insertion and pore-forming ability, αHL has also been used in artificial cell systems to make lipid membranes permeable to small molecules, which can promote internal biochemical reactions like transcription and translation[16]. Multiple seminal studies have demonstrated that αHL can be mutated in several positions without losing its pore forming activity[17].

While αHL assembly leads to spontaneous formation of a trans-membrane protein, only a very small region of each monomer, the loop formed by amino acids 128-131, fully translocates across the lipid membrane upon pore formation (Fig. 1a)[18]. If amino acids 128-131 were modified to contain peptide tags or other small proteins, while still allowing for αHL pore assembly and insertion, artificial cells internally expressing engineered hemolysins would externally display peptide tags that could subsequently interact with the external environment (Fig. 1b). Our envisioned system would constitute a simple way of achieving self-encoded extracellular membrane functionalization in artificial cells.

Several studies have reported mutating single amino acids in αHL loop$_{128-131}$. In particular the G130C-mutant has been shown to be fully active, with the cysteine translocating across the membrane during pore formation[19–23]. However, to our knowledge, there are only a few literature reports about making larger modifications to loop$_{128-131}$[24–27]. It was shown for instance that substituting amino acids 130–134 with 5 histidines does not disrupt the pore forming ability of αHL and endows the pores with the ability to bind zinc ions, which blocks transport[24,25]. Another study replaced amino acid 129 with a cysteine flanked by two flexible linkers, representing an addition of 10 amino acids to the loop[26]. However, it was not investigated if the resulting protein can assemble into functional pores on its own. Instead, hetero-heptamers were formed in which 6 wild-type αHL monomers were mixed with one monomer that had the 10 amino acid insert in the loop. Similarly, it has also been shown that αHL with a peptide sensor in the loop region can form functional heteroheptameric pores with wild type αHL[28,29].

Here we demonstrate that functional peptides up to 52 amino acids in length can be inserted into loop$_{128-131}$ without disrupting αHL's

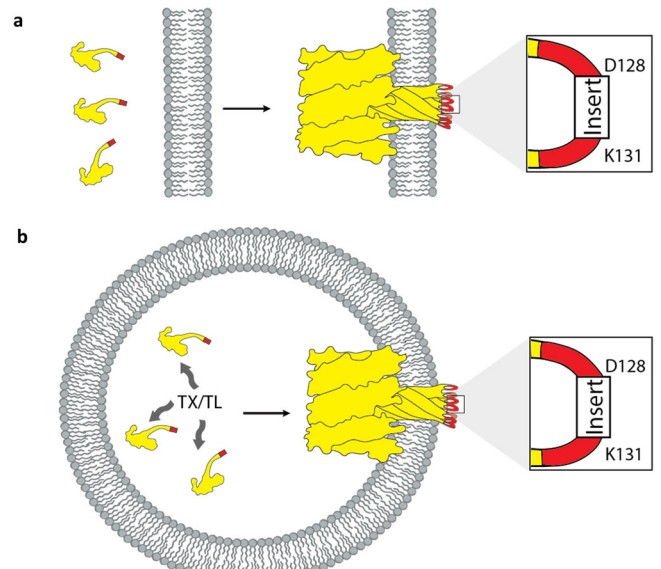

**Fig. 1 | Scheme of α-hemolysin facilitated membrane translocation of peptides. a** αHL translocates peptide inserts in loop$_{128-131}$ across a lipid membrane. If engineered α-hemolysins remain functional, then pore formation and insertion would cause the peptide insert (red) to be translocated across the membrane **b** Intracellular expression of αHL with a peptide insert in loop$_{128-131}$ would enable self-encoded extracellular membrane functionalization in artificial cells.

membrane insertion and pore formation ability. We screen peptides of different sizes to determine the types of inserts that are tolerated. αHL proteins encountering one side of a synthetic membrane can spontaneously translocate peptide epitopes across the membrane and interact with peptide binding antibodies on the other side. Genetically encoding modified αHL that can be expressed within giant unilamellar vesicles (GUVs) leads to the formation of artificial cells that can self-display peptides on their extracellular membrane. By mixing populations of artificial cells that display peptides that interact with one another, we demonstrate self-encoded formation of tissue-like structures. Finally, we show that self-encoded artificial tissue formation is modular and can be combined with other functional systems as demonstrated by implementing a simple artificial cell-cell signaling pathway.

## Results and discussion

### Loop$_{128–131}$ of α-hemolysin tolerates inserts up to around 50 amino acids

We initially tested insertion of a 6XHis-tag between D128 and K131 of the membrane translocating loop of αHL, with the inserted peptide replacing T129 and G130. We also hypothesized that the addition of flexible linkers (L) might make the 6XHis-tag more accessible by increasing the distance between the affinity ligand and the membrane, reducing undesirable steric interactions. A C-terminal GFP fusion was added to enable visualization of membrane binding by the αHL mutant. As a C-terminal fusion, GFP will end up on the opposite side of the membrane as the loop-insert after pore formation. We initially used a simple GUV binding/leakage assay to determine if αHL mutants retained functionality. GUVs encapsulating Cy5 were formed by the inverse emulsion method[30] using a mixture of DOPC and cholesterol. Upon external treatment of Cy5-encapsulating GUVs with αHL containing the 16-mer L-6XHis-L peptide insert (Supplementary Table 1), GFP localization to the membrane was observed as well as leakage of internalized Cy5 dye molecules (Fig. 2), suggesting the formation of a fully functional pore despite the insertion of a 16-mer peptide in the loop$_{128-131}$ region. We next increased the flanking linker to (GGGGS)$_2$

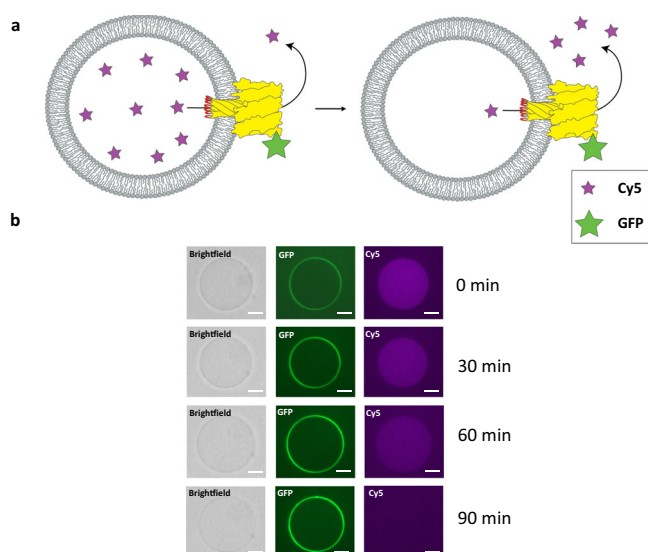

**Fig. 2 | α-hemolysin with a peptide inserted in loop$_{128-131}$ forms functional pores which induce leakage of Cy5 from vesicles. a** General scheme of αHL membrane binding and dye leakage assay. GUVs encapsulating Cy5 (purple stars) are treated with αHL mutants that contain peptide inserts (red) in the loop region. αHL monomers with the respective insert are added to the outside of the GUVs. The αHL is also fused to GFP (green star) to enable visualization of membrane binding. Pore formation in the GUV membrane leads to the leakage of Cy5. **b** Confocal microscopy images of the leakage assay using αHL containing a 16-mer peptide insert (6XHis-tag flanked by two linkers). Green channel shows membrane binding of the αHL-GFP fusion protein. Leakage of Cy5 (purple channel) can be observed over time. Leakage assays were replicated 3 times. Scale bar: 10 μm.

and (GGGGS)$_3$. Surprisingly, despite adding even a 36-mer peptide insert in the loop, GUV binding and leakage assays indicated that the protein was still able to form functional pores (Supplementary Fig. 1).

Encouraged by our results, we next investigated if αHL could tolerate alternative functional peptide inserts. For instance, we were interested if cyclic peptides could be inserted into the loop region, as cyclic peptides have found widespread use in medicinal chemistry[31]. We inserted the sequence for somatostatin-14[32] flanked by two flexible linkers into loop$_{128-131}$. We found that αHL containing a somatostatin-14 insert was also able to induce leakage of Cy5 from GUVs, suggesting that cyclic peptides are tolerated for membrane translocation using αHL loop mutants (Supplementary Fig. 1). Cyclization in somatostatin-14 is a result of disulfide bond formation between two cysteines in the peptide sequence. To confirm that the peptide does form this disulfide bond when inserted into the αHL loop, we used Ellman's reagent to test for free thiols in a sample of αHL with the L-somatostatin-L insert (Supplementary Fig. 2).

To determine the size limitation of the peptide inserts that can be tolerated in the loop region of αHL, we tested a series of additional inserts (Supplementary Tables 1 and 2). We tested a 52 amino acid peptide by inserting a GLP1 peptide hormone[33] flanked by a (GGGGS)$_2$ linker. The GLP1 containing αHL was also able to bind to GUVs and induce Cy5 leakage (Supplementary Fig. 1). However, we found that larger peptide inserts (75–248 amino acids, Supplementary Table 2) generated αHL mutants that were unable to bind to lipid membranes and do not form pores and induce Cy5 leakage from GUVs. Thus, with respect to inserting simple peptides, we estimate that the cutoff length for retaining αHL functionality is approximately 50 amino acids, though undoubtedly the nature of the insert with respect to charge, hydrophobicity, and structure likely have important effects on the assembly of the pore and its interaction with lipid membranes. To better understand how inserts in the loop of the αHL-pore influence small molecule leakage, we also attempted to quantify leakage induced

by each αHL-protein described in this study. By determining Cy5 fluorescence of αHL treated populations of GUVs at different time points, we could show that leakage rate through the αHL pore seems to depend on both insert size and charge. (Supplementary Fig. 3).

While membrane permeabilization is often desired in the construction of artificial cells as it enables small molecule exchange between the artificial cells and the extracellular environment, the data in Supplementary Fig. 3 could be used as a starting point for the design of peptide translocating pores with inhibited leakage characteristics. Adding a long glycine-serine linker for instance, might be a way to slow down leakage and create a more closed off pore.

### Peptide inserts in αHL loop$_{128-131}$ can interact with large biomolecules after membrane translocation

We wanted to confirm that the peptide inserts in the loop get translocated across the membrane and are accessible from the other side. One way to test translocation is to determine if large macromolecules encapsulated in GUVs, like monoclonal antibodies, can recognize and bind the translocated peptide if the αHL loop mutant is delivered from the outside of the GUV. We generated GUVs encapsulating Cy5-conjugated antibodies directed against the various peptides (6XHis, somatostatin, GLP-1) that were shown to insert into αHL without disrupting pore formation. We individually encapsulated the fluorescent antibodies into GUVs. GUVs were then treated with the αHL loop mutant containing the appropriate peptide insert. If the αHL loop mutants fully translocate across the membrane and the loop is fully accessible on the other side, then antibody recruitment to the membrane would take place and this would be visualized by Cy5 fluorescence at the membrane (Fig. 3a). Because monoclonal antibodies have a high molecular weight, they themselves cannot diffuse through the αHL pores, and therefore any observed membrane staining should be due to internal binding[16]. As an initial test, we treated GUVs encapsulating an anti-6XHis antibody conjugated to Cy5 with αHL-GFP containing the L-6XHis-L insert. Using fluorescence microscopy, we observed that, after treatment with the αHL fusion protein, antibodies became localized to the GUV membrane (Fig. 3b). Incubation at room temperature for one hour resulted in nearly all GUVs showing membrane localization of the Cy5 anti-6XHis antibodies (Fig. 3c). In comparison, untreated GUVs encapsulating Cy5-conjugated anti-6XHis antibodies do not show membrane localization of the antibody even after incubation for several hours (Supplementary Fig. 4a). Similarly, GUVs encapsulating Cy5 anti-6XHis-tag antibody show no membrane localization of Cy5-antibody after 4 hours of treatment with αHL L$_2$-GLP1-L$_2$ (Supplementary Fig. 4b). By using different monoclonal antibodies, similar results were obtained for all the peptide inserts we had previously tested, suggesting that our approach is a general method for translocating peptides across lipid membranes such that they are accessible to binding ligands (Fig. 3d). In addition, we also performed a similar experiment where αHL is expressed inside GUVs and the antibody is added to the outside, showing that translocation of the insert is independent of membrane curvature (Supplementary Fig. 5). This is in accordance with the literature, as it has been shown that αHL can insert into membranes of both negative and positive curvature, with a slight preference for membranes of negative curvature[34]. By quantifying the antibody binding data after peptide translocation, we were able to show that the L-6XHis-L insert seems to be able to localize the largest amount of antibody to the membrane. All the other inserts are fairly similar when it comes to antibody binding after membrane translocation despite their difference in amino acid sequence (Fig. 3e).

### Cryo-EM reveals that the peptide inserts in loop$_{128-131}$ of αHL do not interfere with the formation of the heptameric pore structure

To further verify that inserts in loop$_{128-131}$ of αHL do not interfere with pore formation and membrane insertion, we collected structural data

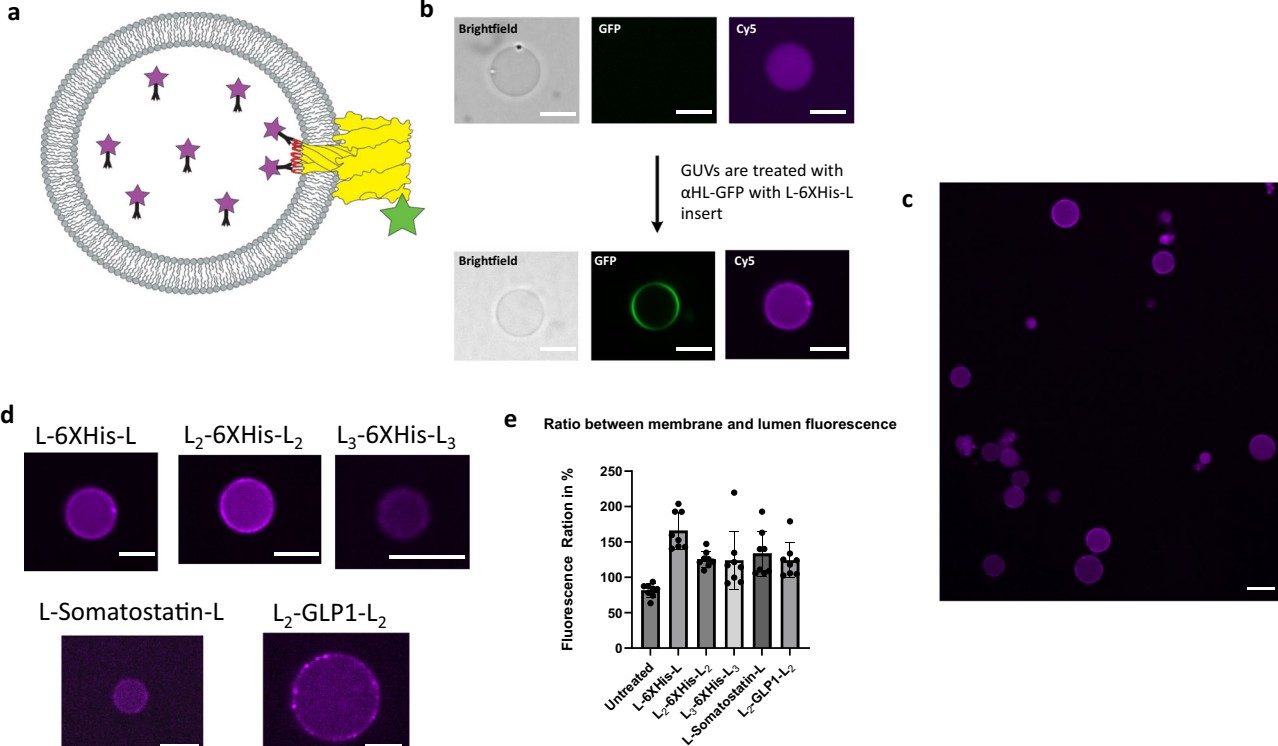

**Fig. 3 | Peptide inserts in loop₁₂₈₋₁₃₁ can interact with encapsulated antibodies after membrane translocation. a** General scheme for testing translocation of peptides by GUV antibody binding assay. A Cy5-conjugated monoclonal antibody specific for a given peptide insert is encapsulated into GUVs. αHL monomers with the respective insert are added to the outside of the GUVs. Upon external treatment of GUVs with αHL-GFP fusion proteins containing the peptide insert, the antibody will localize to the membrane, indicating that the peptide insert has fully translocated across the membrane and can be bound by the internal antibody. **b** GUVs encapsulated with Cy5 modified anti-6XHis-tag antibodies before and after treatment with αHL-GFP containing the L-6XHis-L loop insert. The GUV membrane shows increased Cy5 (antibody) signal after treatment. Scale bar: 10 μm. **c** A larger population of GUVs encapsulating the Cy5 anti-6XHis-tag antibody, all showing increased membrane fluorescence after treatment with αHL-GFP containing the L-

6XHis-L loop insert. Scale bar: 10 μm. **d** Cy5 antibody assays with all the peptide inserts that were shown to form functional pores. Scale bar: 10 μm. **e** Quantitative analysis of the antibody binding assay. The ratio of membrane fluorescence to lumen fluorescence after translocation of the peptides shown in **d** is plotted (further information in Supplementary Fig. 6). Untreated GUVs show a mean fluorescence ratio of 82%, whereas treatment with hemolysin constructs leads to a fluorescence ratio increase of at least 50%. Data ($n = 8$ GUVs per treatment group) was analyzed using an independent $t$-test (two-tailed). Statistical significance was tested for each treatment group against the untreated group. $p < 0.0001$ for L-6XHis-L; $p < 0.0001$ for L₂-6XHis-L₂; $p = 0.0135$ for L₃-6XHis-L₃; $p = 0.0006$ for L-Somatostatin-L; $p = 0.0005$ for L₂-GLP1-L₂. Source data are provided as a Source Data file.

using cryo-EM. We chose to investigate αHL mutants with the L₂-GLP1-L₂ loop insert because it was the largest insert that showed pore formation and peptide translocation based on our previously discussed GUV experiments. Due to its size, the effects of the L₂-GLP1-L₂ insert on αHL protein structure and functionality should be the largest out of all the inserts we tested. Adapting a previously published protocol[35], we prepared small unilamellar vesicles (SUVs) through hydration of a lipid film (DOPC 60%, cholesterol 40%) followed by extrusion through a 100 nm membrane, and then externally treated these vesicles with αHL containing the L₂-GLP1-L₂ insert, which was expressed using the PURExpress® system. Cryo-EM images revealed the formation of numerous αHL pores embedded in the membranes of the SUVs (Fig. 4a). It is possible to see the cap of the αHL pores both from a side-view (red arrow) and from a top-down-view (blue arrow). In addition, we were also able to generate 2D class averages showing a top-view of the heptameric αHL pore as well as a low-resolution side-view of the cap in the membrane (Fig. 4b). By inducing pore formation through the addition of sodium deoxycholate micelles, we were able to generate 2D class averages with a higher resolution (Supplementary Fig. 7). These results further confirm that the loop₁₂₈₋₁₃₁-region of αHL tolerates large peptide inserts without losing its pore forming or membrane insertion abilities.

In addition, we also looked at the effect of peptide inserts on the conductance of the αHL pore in lipid bilayer channel recordings. For

the L-6XHis-L insert (Fig. 4c) we could observe pore blockage events, even in the absence of molecules such as PEG that have been shown to be translocated through the pore[36,37]. We hypothesize that this is due to the flexible nature of the glycine serine linkers, which might allow the insert to flip in and out of the pore leading to these pore blocking events. Previous work has shown that flexible inserts in the cap region can trigger similar pore blockage events during conductance measurements[38]. Consistent with this hypothesis, pores formed from αHL containing longer inserts, like the L₂-GLP1-L₂ insert (Fig. d), showed an increase in the frequency of these pore blocking events. Despite the frequent pore blocking events, we were able to observe an approximate current flow between 0.1 nA and 0.2 nA, in line with our previous experiments demonstrating that loop inserts do not inhibit the ability of αHL to form pores that can induce the leakage of small molecules across lipid membranes.

### α-Hemolysin-peptide fusions enable self-encoded extracellular membrane functionalization of artificial cells and the formation of artificial tissue-like structures

To display peptides on artificial cells using engineered αHL, we encapsulated an in vitro transcription-translation system in a GUV, adapting previously published protocols[39]. Each artificial cell would then be able to functionalize its own extracellular membrane through internal expression of a gene coding for αHL with a functional peptide

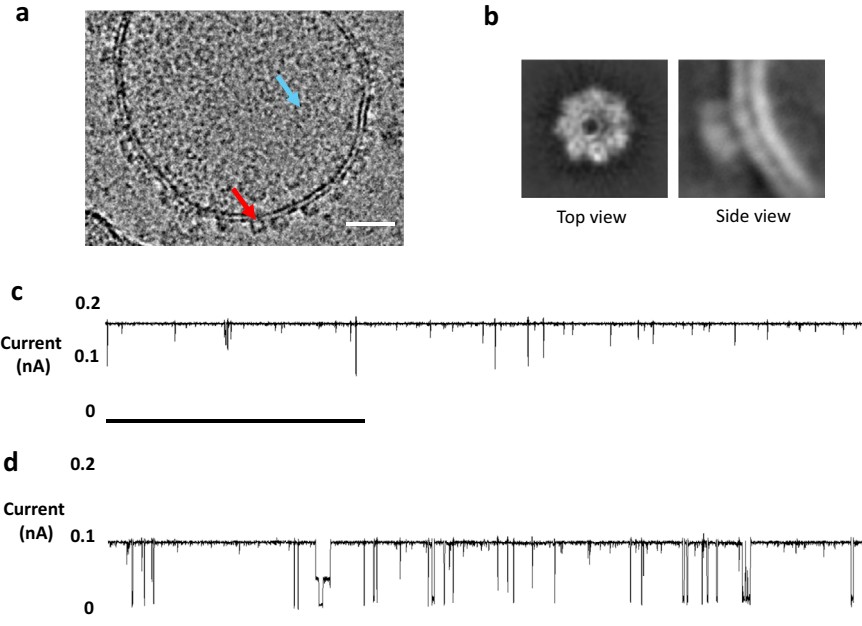

**Fig. 4 | Cryo-EM data and lipid bilayer channel recordings of αHL with peptide inserts in loop$_{128-131}$. a** Cryo-EM image of a representative lipid vesicle after external treatment with αHL containing the L$_2$-GLP1-L$_2$ loop insert. Red arrow: Side-view of αHL cap. Blue arrow: Top view of αHL cap. Scale bar: 10 nm. **b** 2D class averages of cryo-EM images of αHL containing the L$_2$-GLP1-L$_2$ loop insert in a lipid bilayer. Top view shows the heptameric pore structure. **c** Lipid bilayer channel recordings of αHL containing the L-6XHis-L loop insert showing pore blockage events. **d** Lipid bilayer channel recording of αHL containing the L$_2$-GLP1-L$_2$ loop insert. Increasing the length of the flexible linker increases the frequency of pore blockage events. Mean conductance values for αHL with the L-6XHis-L and αHL with the L$_2$-GLP1-L$_2$ are 3.25 nS ± 0.125 nS (50 mV; $n = 3$) and 1.67 nS ± 0.109 nS (50 mV; $n = 3$) respectively. Scale bar: 1 s.

insert in loop$_{128-131}$. Upon expression of the protein inside the membrane-bound artificial cell, the soluble monomer would be expected to assemble into heptameric pores on the lipid membrane, which would lead to spontaneous translocation of 7 copies of the peptide insert across the membrane onto the extracellular membrane side (Fig. 1b).

While several possible applications could be imagined for externally displayed self-encoded peptides, we decided to explore whether we could use αHL loop mutants to encode for the spontaneous generation of synthetic tissues. While engineering artificial cells has been a long-standing goal of bottom-up synthetic biology, self-organization and self-assembly of individual artificial cells into multi-cellular structures may enable significantly expanded functionality and complexity[40–42]. Examples of current ways of implementing synthetic cell-cell adhesion are the use of affinity ligands (e.g. streptavidin and biotin)[43], electrostatic interactions[44], DNA-directed assembly[45], or the reconstitution of natural cellular proteins like claudin[46]. However, one significant difference between all these methods compared to natural cells is that they require the researcher to manually modify the artificial cell membrane externally, for instance by doping the membrane with an affinity ligand. In contrast, natural cells can functionalize their own outer membranes through the self-encoded internal expression and subsequent insertion of membrane proteins which in turn can induce cell-cell adhesion[47]. We envisioned that our described αHL peptide translocation system could be used in artificial cells to overcome this limitation and create a self-encoded cell-cell adhesion system.

To genetically encode self-assembly into artificial cells, we selected two peptide inserts expected to bind to one another. A simple strategy is to design peptide inserts that can associate due to electrostatic interactions (Fig. 5a). We created two αHL mutants with peptide loop inserts K3 and E3 (each 17 amino acids in length, sequences given in Fig. 5a) which would be predicted to interact with each other by forming salt-bridges between the positively charged lysine and the negatively charged glutamate residues[48–50]. We tested the ability of αHL to tolerate each insert by performing a membrane binding and leakage assay as previously described (Supplementary Fig. 8). Each heptameric complex can translocate 21 positive or negative charges across an artificial cell membrane. Complimentary artificial cells bearing opposite charges can interact, eventually leading to cell-cell binding and aggregation of multiple cells into artificial tissue-like structures (Fig. 5b).

We encapsulated the PURExpress® system into GUVs for expression of αHL with either the K3 or E3 loop insert using the T7p14 vector (myTXTL®) as an expression system. The GUVs were formed through the inverse emulsion method with a lipid composition of 60% DOPC and 40% cholesterol. To distinguish GUVs expressing the K3 insert from GUVs expressing the E3 insert, we marked the GUVs by also encapsulating pre-expressed mCherry (K3 insert) or CFP (E3 insert). The two GUV populations were mixed at equal concentrations and protein expression was induced at 37 °C. After one hour of protein expression, the GUVs started to aggregate. The two populations of GUVs showed strong GUV-GUV interactions which ultimately led to the formation of a tissue-like vesicular network structure that had alternating contacts between vesicles displaying the K3 and E3 peptide insert (Fig. 5c). GUV-GUV-interactions were specific to the K3 and E3 inserts. With insufficient protein expression, we observed no formation of aggregates (Supplementary Fig. 9). Control experiments with GUVs only expressing the K3 insert or only expressing the E3 insert also did not lead to the formation of GUV networks (Supplementary Figs. 10 and 11).

## Formation of self-encoded artificial tissue-like structures facilitates a simple signaling pathway

Apart from enabling cell-cell adhesion, self-encoding artificial tissue formation with αHL mutants can provide further functionality due to the formation of pores in the membranes. αHL with the K3 and E3 insert forms functional pores that allow small molecules to leak across membranes (Supplementary Fig. 8). This in turn also means that

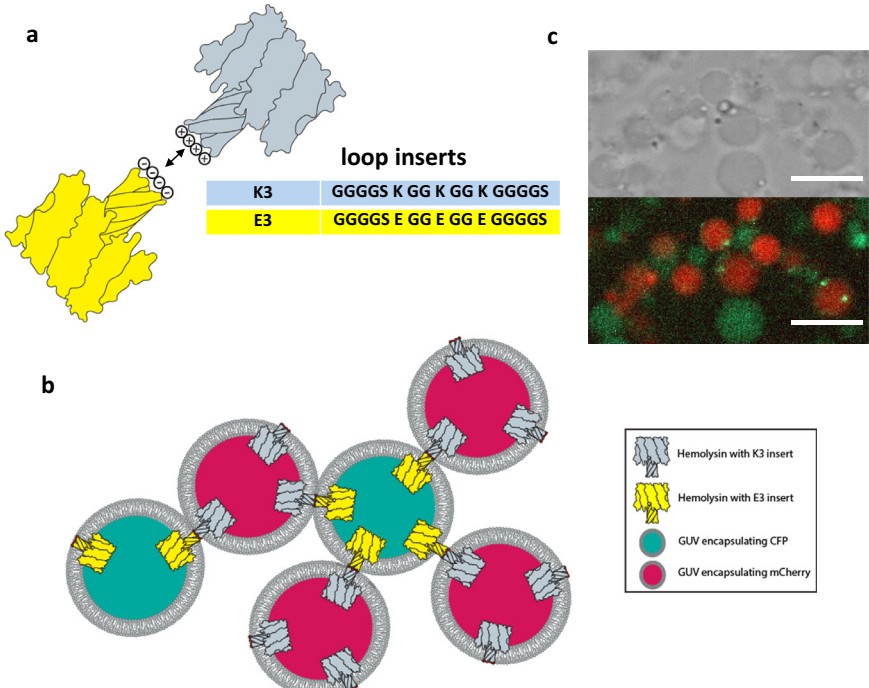

**Fig. 5 | Self-encoded membrane functionalization system enables artificial tissue formation. a** αHL pores with the K3 peptide insert can interact with pores containing the E3 peptide insert due to electrostatic interactions. **b** General scheme of the tissue formation experiment. Two populations of GUVs express αHL with one of two mutually interacting peptide inserts, K3 (positively charged) or E3 (negatively charged). After internal protein expression, αHL oligomerizes on the artificial membrane and forms pores, while at the same time translocating the K3 or E3 peptide insert across the membrane for extracellular display. After translocation, the K3 expressing GUVs can interact with E3 expressing GUVs leading to GUV self-assembly into tissue-like structures. **c** Confocal microscopy image of tissue-like structures formed through mixing K3-αHL and E3-αHL expressing GUVs. Specific interactions between K3-αHL and E3-αHL expressing vesicles were observed and controls (see supporting information) did not result in the formation of assemblies. Scale bar: 10 μm. Shown are representative images from three independent experiments.

vesicles that are part of the artificial tissue are permeable to small molecules. Improved transport of small molecules through nanopores could help facilitate artificial cell-cell signaling within the tissue. While individual artificial cells within a population often act independently of one another, interconnected artificial tissues may be designed to perform specialized actions based on cell-cell communication and interaction. As such, artificial tissues could exceed simple independent and non-interacting artificial cells because communication enables collective action and the formation of spatially ordered tissues, which would be a prerequisite for the creation of life-like structures reminiscent of natural tissues[51]. Previous studies have used covalently fused αHL-pores to enable small molecule exchange in artificial tissues[52]. In our study, we utilize reversible electrostatic interactions at the mouth of the αHL-pore to promote the formation of artificial tissue. Although direct small molecule exchange between two artificial cells through these electrostatically interacting pores might occur, the weaker and reversible nature of electrostatic interactions compared to covalent bonds means that the connected pores are not necessarily closed-off to the extracellular environment.

As a proof of concept, we envisioned engineering a simple signal transduction pathway, where one population of GUVs can generate and send a signaling molecule to a second population of GUVs, which are able to recognize the signaling molecule and produce a detectable readout. We decided on using hydrogen peroxide as the signaling molecule. Hydrogen peroxide has been shown to readily diffuse across lipid membranes[53] and, by adding catalase to the outer solution, hydrogen peroxide that diffuses outside of the artificial cells can be rapidly quenched (Fig. 6a). As a readout, we used the protein Hyper7, a hydrogen peroxide responsive fluorescent protein[54]. To generate hydrogen peroxide we used glucose oxidase[55–57]. One population of GUVs, marked with pre-expressed mCherry, expresses αHL with the K3

loop insert and also contains glucose oxidase (sender cells). A second population of GUVs expresses αHL with the E3 loop insert and contains Hyper7 (receiver cells). Signaling is initiated through the addition of glucose. Expression of the αHL mutants enables tissue formation, based on the interaction of the E3 and K3 insert, and makes the GUVs permeable to glucose, which can traverse the αHL pore. Catalase was added to the outer solution to quench hydrogen peroxide in the extracellular environment. Due to its much higher molecular weight compared to glucose, catalase would not be able to enter the GUVs through the formed αHL pores. Under these experimental conditions we could observe a fluorescence increase for Hyper7 containing GUVs that had assembled with glucose oxidase containing GUVs (Fig. 6b and Fig. 6c). Unaggregated GUVs (marked with a white box) displayed significantly lower Hyper7 fluorescence, indicating that there is proximity and aggregation-based exchange of hydrogen peroxide. A negative control without glucose showed lower fluorescence intensity (Fig. 6c).

In conclusion, we have demonstrated a straightforward method for the self-encoded extracellular functionalization of artificial cells using internally expressed proteins. Our method relies on the insertion of functional peptides into loop$_{128-131}$ of the self-translocating pore toxin αHL. We show for the first time that lengthy peptides, up to approximately 50 amino acids, can be inserted into the αHL loop without affecting its ability to form homoheptameric pores. We also validate by cryo-EM that the inserts do not affect formation of the heptameric pore. Current methods typically achieve artificial cell interactions by chemically modifying the outer membranes of giant vesicles[40,41]. This contrasts with our study, where we have genetically encoded membrane functionalization. The ability of individual artificial cells to encode their own extracellular membrane modification opens ways for programming interactions between artificial cells and their environment. We demonstrate this by programming the

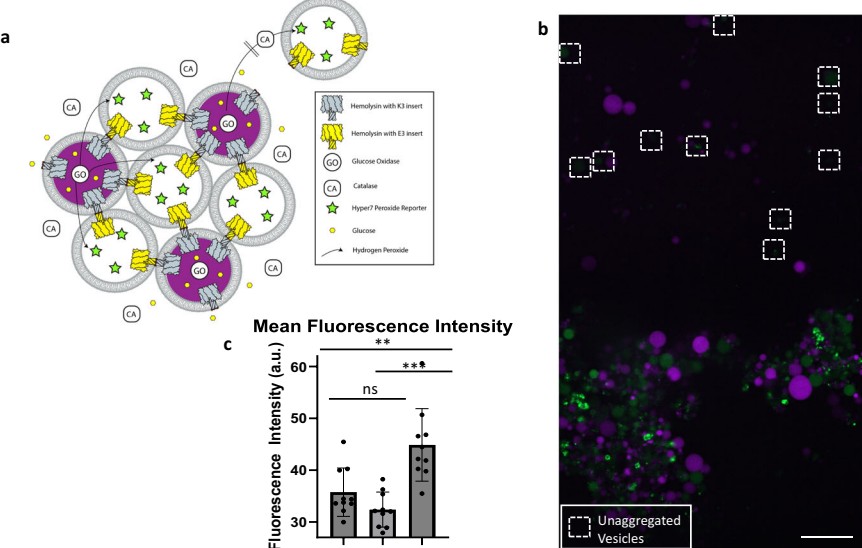

**Fig. 6 | Hydrogen peroxide signaling in self-encoded artificial tissue-like structures. a** General scheme of an artificial cell signaling pathway using hydrogen peroxide. Glucose oxidase (GO) containing sender artificial cells (marked with mCherry) create hydrogen peroxide in the presence of glucose. Hydrogen peroxide can enter receiver artificial cells which contain Hyper7 as a peroxide reporter. Catalase (CA) is added to the extracellular solution to prevent peroxide signaling outside of the artificial tissue-like structures. **b** Confocal microscopy image of hydrogen peroxide signaling in a large tissue-like assembly. Hyper7 containing artificial cells inside the vesicular assembly show increased fluorescence in the green channel. Hyper 7 containing artificial cells outside of tissue (marked with white boxes) show much weaker fluorescence signal. Scale bar: 25 μm.

**c** Quantification of Hyper7 fluorescence intensity. 1: Control experiment. Mean fluorescence of artificial cells in the tissue-like structure in the absence of glucose (Supplementary Fig. 12). 2: Mean fluorescence of artificial cells outside of tissue-like structure in the presence of glucose. 3: Mean fluorescence of artificial cells in the tissue-like structure in the presence of glucose. Error bars represent SD. $n = 10$ GUVs for each group. Statistically significant differences in Fig. 6c are indicated based on an independent $t$-test (two-tailed). 1 vs 2: $p = 0.0830$; 1 vs 3 $p = 0.00351$; 2 vs 3 $p = 0.000210$. Source data are provided as a Source Data file. **b** with increased brightness in the green channel to better visualize GUVs in the white boxes is shown in Supplementary Fig. 13.

formation of functional synthetic tissues using GUVs that self-encode the surface display of interacting peptides.

Our work may also have implications beyond providing a platform for the programmable functionalization of artificial cells. For instance, at the origin of life, it is unlikely there were highly sophisticated translocase systems, and our work may shed light on the mechanisms by which protocells functionalized their membranes and communicated with their environment. Furthermore, a better understanding of membrane translocation could lead to the development of tools for the delivery of macromolecular therapeutics across lipid membranes and into living cells.

## Methods

### General Information
**Lipids.** 1,2-dioleoyl-sn-glycero-3-phosphocholine (18:1 PC; DOPC) (Cat# 850375) was purchased from Avanti Polar Lipids in chloroform. Cholesterol (Cat# 700000P) was also purchased from Avanti Polar Lipids.

**Protein expression and purification reagents.** PURExpress® kit was purchased from NEB (Cat# E6800), myTXTL® T7 Expression Kit from Daicel Arbor Biosciences (Cat# 505024). BL21(DE3) cells were purchased from NEB (Cat# C2527). HisPur™ Ni-NTA Resin was purchased from ThermoFisher (Cat# 88221). PD SpinTrap G-25 columns were purchased from Cytiva (Cat# 28918004).

**Antibodies.** DyLight™ 650-Anti-6XHis-tag antibody was purchased from ThermoFisher (Cat# MA1-21315-D650).

CF™ 647-Anti-Somatostatin antibody was purchased from biorbyt (Cat# orb500667-CF647).

AF® 647-Anti-GLP1 antibody was purchased from Bioss (Cat# bsm-0933M-A647).

Ficoll® 400 was purchased from Millipore Sigma (Cat# F8016). Mineral oil heavy from Fisher Scientific (O122-1) and BSA (Bovine serum albumin) was purchased from Cytiva (SH30574.01). Glucose Oxidase (Cat# G2133) as well as catalase (Cat# C9322) were purchased from Millipore Sigma. Confocal microscopy was done on a Zeiss Cell Observer® SD consisting of a Yokagawa spinning disk system (Yokagawa, Japan) built around an Axio Observer Z1 (Zeiss, Germany) motorized inverted microscope. Microscopy images were analyzed in Fiji/ImageJ. Fluorescence values of individual GUVs are mean fluorescence values generated by manually selecting GUVs in ImageJ with the oval selection tool. For quantification, at least 10 GUVs from each population were averaged.

**Protein expression and purification.** Gene constructs were cloned into the pTNT™ vector (Promega). BL21(DE3) Competent *E. coli* cells (NEB) were transformed with the respective plasmid. 300 ml of LB-medium with carbenicillin (100 μg/ml) was inoculated with a single colony and incubated on a shaker at 37 °C overnight. Cells were collected through centrifugation for 5 min at 5000 g. The resulting cell pellet was resuspended in 3 ml lysis buffer (HEPES 20 mM pH = 7.5, NaCl 500 mM, imidazole 10 mM, PMSF 1 mM). Cells were lysed by sonication on ice (3 s on, 3 s off, 10 min, 50% amplitude). The lysate was centrifuged (15,000 x g, 30 min, 4 °C) and filtered through a 0.45 μm syringe filter. The resulting cleared solution was applied to a gravity Ni-NTA column, loaded with 1 ml HisPur™ Ni-NTA Resin that had been pre-equilibrated with lysis buffer. After incubation with the lysate on an overhead spinner at 4 °C for 30 min, the column was washed 4 times with wash buffer (HEPES 20 mM pH = 7.5, NaCl 500 mM, imidazole 30 mM; 1 ml each wash). Protein was eluted into 4 volumes of elution buffer (HEPES 20 mM pH = 7.5, NaCl 500 mM, imidazole 300 mM; 0.5 ml each elution fraction). Protein fractions were identified by SDS−PAGE and the relevant fractions pooled and dialyzed against

storage buffer (HEPES 20 mM pH = 7.5, NaCl 500 mM). Proteins were stored at 4 °C for several days or at −80 °C for long-term storage. In all experiments, αHL monomers were used without preassembling them into pores.

**Expression of Hyper7.** Hyper7[47] was expressed using the myTXTL® T7 Expression Kit. Expression was set up according to the manufacturer's instructions. After 18 h of protein expression at 29 °C, TCEP was added to a final concentration of 50 mM. Following incubation at RT for 30 min, the solution was centrifuged at 80,000 × *g*, 30 min, 4 °C. Excess TCEP was removed from the supernatant using a PD SpinTrap G-25 column. The resulting reduced protein was concentrated on an Amicon ultra spin-concentrator (10 kDa cut-off) and was used without further purification.

### GUV formation
GUVs were formed through the inverse emulsion method[30]. Briefly, a 1 mM lipid stock (DOPC 60 mol%, cholesterol 40 mol%) in mineral oil was prepared. To 100 µl of this lipid solution was added 10 µl of encapsulation solution, containing all the components to be encapsulated into GUVs. This mixture was emulsified by vortexing. The resulting emulsion was layered on top of lower buffer solution. If not specified otherwise, this lower buffer solution comprised of 20 mM HEPES pH = 7.0, NaCl 500 mM. This two-layer system was subjected to centrifugation at 10,000 × *g* for 10 min. The oil layer was removed and the GUV pellet was resuspended in 20 µl lower buffer. For forming GUVs encapsulating a PURExpress® expression system, this GUV formation protocol was scaled down by half: 50 µl lipid solution, 5 µl encapsulation solution, 50 µl lower buffer solution.

### Vesicle leakage assay
GUVs were formed as described with encapsulation solution containing Cy5 (1 µM) and 3.5% (wt/vol) Ficoll® 400 in buffer (20 mM HEPES pH = 7.0, NaCl 500 mM). The resulting GUVs were treated with protein to a final protein concentration of 10 µM. After incubation on an overhead spinner at room temperature for 2 h, the GUVs were imaged on a confocal microscope.

### Antibody assay
GUVs were formed as described with the encapsulation solution containing a solution of Cy5-conjugated antibody (1:500 dilution) and 3.5% (wt/vol) Ficoll® 400 in buffer (20 mM HEPES pH = 7.0, NaCl 500 mM). The resulting GUVs were treated with protein to a final protein concentration of 10 µM. After incubation at room temperature for 1 h, the GUVs were imaged on a confocal microscope.

### Cryo-EM of αHL pores in lipid bilayers
A 2.5 mM vesicle solution (DOPC 60 mol%, cholesterol 40 mol%) was prepared through hydration of a lipid film with buffer. (Tris 20 mM pH = 7.0, NaCl 250 mM). The resulting vesicles were extruded 11 times through a 100 nm polycarbonate membrane using a commercial mini-extruder (*Avanti* Polar Lipids).

αHL monomer with the L₂-GLP1-L₂ insert was prepared through in-vitro expression with the PURExpress® system. A 25 µl reaction was incubated at 37 °C for 4 h and then purified using reverse His-tag purification according to the manufacturer's instructions. To the resulting protein solution was added the solution of extruded vesicles described above to a final lipid concentration of 1 mM. After incubation at room temperature for 1 h, cryo-EM grids were prepared.

Cryo-electron microscopy grids (Lacey Carbon Film, Electron Microscopy Sciences #LC300-Cu) were glow-discharged (Emitech K350 unit at 20 mA for 30 s), deposited with 3.0 µL of the protein-treated vesicles, blotted for 4 sec, and then plunged into liquid ethane using a Vitrobot (Mark IV, Thermo Fisher Scientific). Images were acquired on a Talos Arctica (FEI) operated at 200 kV equipped with a

Falcon 4i Direct Electron Detector (Thermo Fisher) and collected with a total dose of 40 e/Å2 at 0.95 Å/pixel and at −3 µm nominal defocus. 2837 individual exposures were collected automatically using EPU (Thermo Fisher). Exposures were analyzed in cryoSPARC. Top View 2D class average: Initial particle picks were obtained using cryoSPARC Live's blob picker (100–200 Å circular blobs). The resulting 2247 particles were used for the generation of 2D class averages (20 classes in total). Side View 2D class average: Initial particle picks were obtained using cryoSPARC's manual picker (256 px box size). The resulting 355 particles were used for the generation of 2D class averages (20 classes in total).

### Cryo-EM of detergent-stabilized αHL pores
α-hemolyisn monomer with the L₂-GLP1-L₂ insert was prepared through in-vitro expression with the PURExpress® system. 5 reactions (25 µl each) were incubated at 37 °C for 4 h and then purified using reverse His-tag purification according to the manufacturer's instructions. The resulting protein solution was concentrated on an Amicon ultra spin-concentrator (10 kDa molecular weight cut-off). To induce pore-formation, sodium deoxycholate was added to a concentration of 6.25 mM. After incubation at RT for 30 min, the solution was diluted with buffer (Tris 20 mM pH = 7.0, NaCl = 250 mM) to a sodium deoxycholate concentration of 1 mM. The resulting dilute solution of αHL pores was concentrated using an Amicon ultra spin-concentrator (100 kDa molecular weight cut-off).

All samples were prepared on UltraAuFoil 1.2/1.3, 300 mesh grids that had been freshly plasma-cleaned using a Gatan Solarus II plasma cleaner (10 s, 15 Watts, 75% Ar/25% O₂ atmosphere), deposited with 3.0 µL of the protein solution, blotted for 4 sec, and then plunged into liquid ethane using a Vitrobot (Mark IV, Thermo Fisher Scientific). Images were acquired on a Titan Krios G4 (Thermo Fisher) operated at 300 kV and equipped with a Selectris X energy filter and a Falcon 4 Direct Electron Detector. Micrographs were collected with a total dose of 55 e/Å2 at 0.935 Å/pixel and at −3 to −1 µm nominal defocus range. 5357 individual exposures were collected automatically using EPU (Thermo Fisher). Exposures were analyzed in cryoSPARC. Initial particle picks were obtained using cryoSPARC Live's blob picker (100–200 Å circular blobs), which was used to generate templates for one round of template picking. The resulting 36740 particles were used for the generation of 2D class averages (30 classes in total).

### Single-channel recordings
Recordings were taken on an Orbit Mini instrument (Nanion Technologies) using MECA 4 chips with a 50 µm microcavity. The signal was filtered as such: Range: 2 nA; Sampling frequency (SR): 5 kHz; Final Bandwidth: SR/8. The Orbit Mini was used as described in the manufacturer's instructions. In short: Recording buffer (3 M KCl, 20 mM Tris pH = 7.0, 150 µl) was added to the chip. Lipid membranes were painted from a DOPC solution in n-octane (5 M). 20 µl of a PURExpress expression solution of the respective protein was added to the chip. Measurements were taken at +50 mV.

### Artificial tissue formation
GUVs expressing either αHL with the K3 insert or the E3 insert were formed as described with the encapsulation solution containing a PURExpress® expression mix prepared according to the manufacturer's instructions. Two different batches of GUVs were prepared. For GUVs expressing αHL with the K3 insert, to 5 µl PURExpress® mix was added mCherry to a final concentration of 1 µM, plasmid coding for αHL with the K3 insert (75 ng), and 3.5% (wt/vol) Ficoll® 400. For GUVs expressing αHL with the E3 insert, to 5 µl PURExpress® mix was added CFP to a final concentration of 1 µM, plasmid coding for αHL with the E3 insert (75 ng), and 3.5% (wt/vol) Ficoll® 400. To account for the high protein concentration in the PURExpress® mix, the lower

buffer solution was changed to 50 mM Tris pH = 7.0, alanine 100 mM, BSA (66 μM). The GUV pellet was resuspended in expression buffer (PURExpress® solution A (2.0 μl), H$_2$O (3.0 μl)).

GUVs expressing αHL with the K3 insert and GUVs expressing αHL with the E3 insert were mixed at a ratio of 1:1 and incubated on an overhead rotator at 37 °C for 2 h to induce protein expression and subsequently aggregation into tissue-like structures. The resulting tissue-like structures were imaged by confocal microscopy.

### Hydrogen peroxide signaling in artificial tissues

For the formation of sender and receiver cells the above protocol for artificial tissue formation was modified slightly: For sender GUVs, to 5 μl PURExpress® mix was added mCherry to a final concentration of 1 μM, plasmid coding for αHL with the K3 insert (75 ng), glucose oxidase (1 U/ml) and 3.5% (wt/vol) Ficoll® 400. For receiver GUVs, to 5 μl PURExpress® mix was added plasmid coding for αHL with the E3 insert (75 ng), 3.5% (wt/vol) Ficoll® 400 and Hyper7 to a final concentration of 2 μM. A 1:1 mixture of sender and receiver GUVs were supplemented with 1% (wt/vol) glucose and catalase to a final concentration of 3 U/ml. The resulting mixture was incubated on an overhead rotator at 37 °C for 2 h to induce protein expression, aggregation into tissue-like structures and hydrogen peroxide signaling. The formed tissue-like structures were imaged by confocal microscopy.

### Statistics and reproducibility

Statistical analysis was done using either GraphPad Prism 10 or Microsoft Excel (Version 2409). All relevant information on statistical analysis, including $p$-values, the number of GUVs that were analyzed and the statistical test used can be found in the respective figure legends. No statistical method was used to predetermine sample size. No data were excluded from the analyses and experiments were not randomized. The investigators were not blinded to allocation during experiments and outcome assessment.

### Reporting summary

Further information on research design is available in the Nature Portfolio Reporting Summary linked to this article.

## Data availability

All data supporting the findings of this study are available within the main text and supplementary information. Source data are provided with this paper.

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

## Acknowledgements

This material is based upon work supported by the Department of Defense through the Vannevar Bush Faculty Fellowship (Award N00014-22-1-2800, N.K.D.). The authors acknowledge the facilities along with the scientific and technical assistance of the staff of the cryo-EM facility at UC San Diego, in particular Dr. Mariusz Matyszewski and Dr. Inga Kuschnerus.

## Author contributions

Conceptualization: A.H. and N.K.D. Investigation: A.H. and A.F. Funding acquisition: N.K.D. Supervision: N.K.D. Writing & editing: A.H., A.F., N.K.D.

## Competing interests

A.H. and N.K.D. are inventors on an international patent application (PCT/US2024/015069, published as WO2024168197A2), related to the use of α-hemolysin for translocation of cargos across membranes. A.F. has no competing interests to declare.
