## [Transparent Peer Review file · Nature Communications]

Encoding extracellular modification of artificial cell membranes using engineered self-translocating proteins

Corresponding Author: Professor Neal Devaraj

Version 0:

Reviewer comments:

Reviewer #1

(Remarks to the Author)

Harjun et al. describes a new way of permeabilizing lipid vesicles and modifying their surfaces with various peptides. At the heart of this concept lies the engineering of the pore forming protein toxin, alpha-HL, where peptides are inserted in an outer membrane loop. Through cell free protein expression, the engineered alpha-HL is produced inside lipid vesicles, inserts into the membrane with the introduced peptide, resulting in permeabilization and presentation of the inserted peptide on the extracellular side. While the overall concept has its merits the experimental evidence for the realization and its use is limited. In most cases, the engineered alpha-HL is added externally as purified protein to the vesicles, such that the engineered peptide is not on the desired extracellular side but oriented into the vesicle. There is only one set of experiments where the peptide containing alpha-HL is produced through cell free protein expression in the vesicles and is exposed to the extracellular side with only a 17 amino acid sequence. Another weakness of the approach is that the presentation of the peptide is always coupled to pore formation, which results in membrane permeabilization. At present this work seems premature for publication. Below are some suggestions for improvement of the manuscript.

1) The study draws most of its conclusions from images of a single giant unilamellar vesicle (GUV), as depicted in Figures 2 and 3. However, it's crucial to acknowledge the inherent variability within vesicle samples, encompassing differences in size, membrane defects, lamellarity, and permeability due to their bulk preparation, as evidenced by supporting information images. These variations can significantly impact both the insertion of proteins into the membrane and their permeability. Hence, conducting quantitative image analysis across a statistically significant number of GUVs or population level analysis with FACS are essential for supporting the claims of the study. This applies to investigating both alpha-HL insertion efficiency and vesicle permeabilization upon pore formation. In addition, a negative control in the absence of alpha-HL is missing. Its inclusion is vital to ensuring that Cy5 does not leak out of the vesicles over time.

2) In the study different peptides with different lengths and sequence (Ln-6XHis-Ln, L-Somatostatin-L, L2-GLP1-L2) are investigated for their efficacy to insert into the lipid membrane and result in permeabilization. Currently their performance is only described qualitatively making it impossible to compare the various peptides. Monitoring time profiles of alpha-HL insertion and quantifying population level permeabilization would guide future users in their choice of sequence.

3) In the study there is only one example of the peptides (two sequences of 17 amino acids) being expressed inside the vesicles and their translocation to the extracellular side. In all the other examples, especially with the larger peptides inserts in the membrane in the opposite orientation, which is contrary to the aim of the study. To prove the general concept also advertised in the title, more and convincing applications of the correct extracellular modification of artificial cell membranes are needed.

4) The application of a bioactive extracellular peptide on the artificial membrane lacks a compelling demonstration. Initially, constructs featuring interesting peptides like somatostatin and GLP1 are developed and their subsequent utilization to modify cell behavior would significantly broaden the scope and enhance the overall impact this new technology.

5) From the data in Figure 4c-d, the authors come to the conclusion that the longer and more flexible peptide (Figure 4d) increases the frequency of pore blockage compared to the short peptide (Figure 4c). However, the curves suggest the opposite as in Figure 4c there are more low current events compared to in Figure 4d where the current is constantly high throughout the measurement.

6) In Figure 5 a control in the absence of the alpha-HL pores that shown background level signal from the Hyper-7 sensor is missing. In addition, the analysis method and the statistics of the quantification are not described.

7) The manuscript is missing a conclusion that contextualizes the results within the existing literature.

Minor point:

1) Fig. 1 gives the false impression that in general the alpha-HL is expressed inside the vesicles. For the initial experiments in Fig. 2-4 the figures should illustrate that alpha-HL is added as purified protein externally and that the designed peptide is presented into the lumen of the vesicle. At the same time this should be clearly stated in the text.

2) The authors should clarify that the C-terminal GFP is positioned on the opposite side of the membrane to the inserted peptide.

Reviewer #2

(Remarks to the Author)

The authors use the ability of the bacterial pore-forming protein alpha-hemolysin to translocate a segment of its own polypeptide chain across lipid bilayers (as part of the pore assembly process) to move polypeptide sequences into the interior or to the surface of giant unilamellar vesicles (GUVs). This is not new, but well worth further exploration. In addition, they explore the ability of the translocated chains to connect and form network-like structures from GUVs, which is new and an intriguing means to form tissue-like materials. However, the paper needs revision to bolster the data quality and scientific presentation, and thereby substantiate the conclusions.

The Methods section must be improved to ensure reproducibility and the validity of conclusions. Information is missing, for example, on the sequences of the protein constructs that have been used, and whether monomers or heptamers of alpha-hemolysin were used in the various experiments. Evidence should be provided for the presence of a disulfide bond in the somatostatin-containing mutant. Quantitative analysis of the fluorescence changes over time during both the dye leakage and the antibody-nanopore co-localisation experiments would be informative. The current traces in Fig. 4 are not convincing with substantial fluctuations in the baselines. Only one trace has been shown for each mutant, i.e. it is not clear whether experiments were repeated. Statistically significant mean conductance values for each mutant should be reported.

Respect should be paid to prior literature. For example, the translocation of long segments of polypeptide chain by the same mechanism as that exploited by the authors and access of the chain to an enzyme in the recipient compartment was demonstrated a decade ago (L. Harrington et al. PNAS, 110, E4417-26 (2013). L. Harrington et al., ACIE, 54, 8154–8159 (2015)).

The aggregation of vesicles to form tissue-like materials has also been investigated previously. One approach uses an electrostatic mechanism related to that explored by the authors; 'colonies' of GUVs with negative surface charge were produced by aggregation with poly-L-arginine (P. Carrara et al. ChemBiochem 13, 1497–1502 (2012), as cited by the authors). A more sophisticated approach is DNA-directed assembly (M. Hadorn et al. Langmuir 32, 3561–3566 (2016)).

Finally, the authors might comment on whether they think molecules can diffuse from one GUV to another through two interacting nanopores containing loops of opposite charge, an idea which also has been explored previously (S. Mantri, et al. Nature Commun 4, 1725 (2013)).

Version 1:

Reviewer comments:

Reviewer #1

(Remarks to the Author)

In the revisions my concerns were only partially addressed and the manuscript was only marginally improved. As also pointed out by Reviewer 2, the formation of prototissues based on electrostatic interactions is not a convincing application of the system. The same thing has already been achieved with lipids of opposite charge and the current system is a more complicated way of doing this. The value of modifying the GUVs with peptides will come from using bioactive peptides that can engage specific cell receptors. The authors argue that the application with cellularly active peptides would require more optimization but this is an important point if this system is going to be used by others. In the current form the work lacks in novelty to be published in Nature Communications and after addressing the more technical concerns the manuscript could be published in a more specialized journal such as Communications Biology.

The authors point out that permeabilization is a desired feature but just as well may become problem smaller components dilute into the environment.

The data in Supplementary Fig. 3 should be moved to the main text as it provides important information on the performance of the system. The statistical significance of this data needs to be verified. The error bars in this figure are very large and the GUVs seem to be leaking even in the absence of alpha-HL.

The manuscript has numerous problems concerning the labeling of the figures. The figure legends don't match with the results and don't provide sufficient details to understand the results. For example, Supplementary Fig. 1 does not show the leakage of Cy5 from the GUVs over time and the time point at which the data is reported is not mentioned. A full timeline, with the initial GUV and the change over time must be provided. All figures should be reviewed with care.

The quantification of the data in Figure 3 is also still missing. Currently, the conclusions still rely on single images.

I appreciate the demonstration of the in vitro protein synthesis in a second set of experiments. This data should be moved to the main text.

Figure 6 needs to be revised. The graph is missing its y-axis label. The data analysis and statistical analysis are not clear. The GUVs that are claimed to be in the white square are not visible in any image. I would recommend using a membrane dye to clearly visualize the GUVs. The samples that are labeled as 1/2/3 in the graph are not described in the text.

The material and methods are not sufficiently detailed for others to reproduce the data. The image analysis routine for the permeability assay and the communication is not clear.

Reviewer #2

(Remarks to the Author)

The authors have made an unusually strong and largely successful attempt to reply to the points raised by both Reviewers. The outcome is a high-quality piece of work with interesting implications in synthetic biology, drug delivery into cells, and more.

This Reviewer has a few remaining comments, that could be left to the authors to consider:

1. Abstract and title inflation should be controlled at the editorial level; they are all some readers look at. The title here is fine. However, the Abstract does not make it clear that the basis of the work is an elaboration of a previous finding: that aHL can translocate fused sequences across bilayers. The reported work is certainly very interesting indeed, but it is incorrect to imply that a new property of aHL itself has been discovered.

Second, the authors end the Abstract by stating that they have made "artificial tissue like structures capable of signal transduction". This last phrase implies, subtly, that the engineered pores are involved in signaling, which they are not. Presumably any means of clustering the vesicles would work because the H₂O₂ likely passes through the external solution and enters the receiver "cells" before dilution as indicated in Figure 6.

2. The differences in dye leakage from GUVs caused by the various mutants is not considered in detail. We note first that monomers were used. Considerations include:

(i) the binding rate and affinity of aHL to the GUVs. This is especially important because an active heptamer must be formed; (ii) the lag period, which can be many minutes depending on the conditions and arises from (a) prepore assembly (b) insertion of the prepore. In the paper, the most obvious lag is in Figure S3c; (iii) the rate of transport once a membrane-spanning heptamer has formed. This depends on the nature of the transported molecule and cannot simply be estimated from the unitary ionic conductance (e.g. Wu, Y., et al. Permeation of styryl dyes through nanometer-scale pores in membranes. *Biochemistry* 50, 7493-7502 (2011)).

Reviewer #1 (Remarks to the Author):

Comment 1:

While the overall concept has its merits the experimental evidence for the realization and its use is limited. In most cases, the engineered alfa-HL is added externally as purified protein to the vesicles, such that the engineered peptide is not on the desired extracellular side but oriented into the vesicle. There is only one set of experiments where the peptide containing alfa-HL is produced through cell free protein expression in the vesicles and is exposed to the extracellular side with only a 17 amino acid sequence.

Response: We thank the reviewer for this comment. We have now included an additional experiment where hemolysin is expressed inside GUVs. To demonstrate that translocation works in both directions, we used aHL with the largest insert (L₂-GLP1-L₂) tested in our work. aHL with the L₂-GLP1-L₂ insert was expressed inside GUVs using PURExpress. After expression, Cy5-anti-GLP1 antibody was added to the outside of the GUVs. The antibody binds the GUV. We also performed a control reaction in which we added a Cy5-anti-His tag antibody to GUVs without aHL, which did not lead to antibody binding to the membrane, indicating that there is no unspecific binding of antibody to the GUV membrane (Supplementary Fig. 5). Together our data demonstrates that the aHL with GLP1 insert can be translocated both from outside to inside and inside to outside.

Comment 2:

Another weakness of the approach is that the presentation of the peptide is always coupled to pore formation, which results in membrane permeabilization.

Response: We thank the reviewer for pointing this out and have added a sentence to the manuscript making it clear that our approach results in membrane permeabilization. That being said, we wish to point out that membrane permeabilization is often desired in the construction of artificial cells to enable the transport of small molecules into and out of the vesicle. Such transport has been used in past work to improve internal protein translation (see *Proc. Natl. Acad. Sci. USA* 101, 51, (2004): 17669-17674). Furthermore, the permeabilization we observe appears to be controllable by the nature of the linker sequence, and future work may be able to use the inserts to gate membrane permeability. We have added a few sentences to the manuscript discussing these points in more detail.

Comment 3:

The study draws most of its conclusions from images of a single giant unilamellar vesicle (GUV), as depicted in Figures 2 and 3. However, it's crucial to acknowledge the inherent variability within vesicle samples, encompassing differences in size, membrane defects, lamellarity, and permeability due to their bulk preparation, as evidenced by supporting information images. These variations can significantly impact both the insertion of proteins into the membrane and their permeability. Hence, conducting quantitative image analysis across a statistically significant number of GUVs or population level analysis with FACS are essential for supporting the claims of the study. This applies to investigating both alfa-HL insertion efficiency and vesicle permeabilization upon pore formation. In addition, a negative control in the absence of alfa-HL is missing. Its inclusion is vital to ensuring that Cy5 does not leak out of the vesicles over time.

Response: We thank the reviewer for the comments and suggestions. As per the reviewer's suggestion we quantified the leakage assay using image analysis. For each of the engineered hemolysins described in the manuscript we did the following experiment: Cy5 encapsulating GUVs were treated with aHL-GFP and fluorescence was measured over time. Every 45 minutes, a population of GUVs ($n > 10$) was imaged and analyzed for Cy5 fluorescence. Mean fluorescence intensity at different timepoints was calculated for each hemolysin mutant and for untreated negative control GUVs (Supplementary Fig. 3). As can be seen, negative controls have negligible leakage of Cy5 over time.

Comment 4:

In the study different peptides with different lengths and sequence (Ln-6XHis-Ln, L-Somatostatin-L, L2-GLP1-L2) are investigated for their efficacy to insert into the lipid membrane and result in permeabilization. Currently their performance is only described qualitatively making it impossible to compare the various peptides. Monitoring time profiles of alpha-HL insertion and quantifying population level permeabilization would guide future users in their choice of sequence.

Response: We thank the reviewer for this comment and agree this is an important question. We have performed additional experiments quantifying permeabilization and have added the data in Supplementary Fig. 3. By comparing Cy5 leakage induced by the different aHL mutants described in this study, we could show that insert length affects leakage rate. For instance, with the 6XHis-tag inserts, increased linker length leads to slower leakage of Cy5. At the same time, insert size is not the only factor influencing leakage. We could observe for instance that charge plays an important role. aHL with the K3 insert induces leakage much faster than aHL with the E3 insert. We hypothesize that this might be due to electrostatic interactions of the negatively charged loop of the E3 pore with the negatively charged Cy5 dye, slowing down leakage. We agree that this could guide future users in designing sequences for translocation. aHL insertion happens nearly instantaneously so creating an insertion time profile would be very challenging. In our experience, in the time it takes between treatment of the vesicles and the start of imaging on the microscope, insertion has already taken place for all the engineered hemolysins described in this study.

Comment 5:

In the study there is only one example of the peptides (two sequences of 17 amino acids) being expressed inside the vesicles and their translocation to the extracellular side. In all the other examples, especially with the larger peptides inserts in the membrane in the opposite orientation, which is contrary to the aim of the study. To prove the general concept also advertised in the title, more and convincing applications of the correct extracellular modification of artificial cell membranes are needed.

Response: In order to further show that our concept works independent of orientation we repeated the antibody binding assay for aHL with largest functional insert described in our manuscript. GLP1 peptide can be translocated by aHL from the lumen of a GUV to the outside where it can be bound by a fluorescent GLP1 antibody and have added this data to the manuscript along with a brief explanation (Supplementary Fig. 5) Our findings are in accordance with the literature as it is known aHL can insert in

membranes with both negative and positive curvature, with a slight preference for negative curvature (*ACS Chem. Biol.* 2015, 10, 7, 1694–1701). We believe the past literature combined with our previous and additional experiments supports that the directionality of membrane translocation should not matter significantly with respect to aHL insertion. We have added a sentence in the manuscript to make this clearer along with the relevant citation.

Comment 6:

The application of a bioactive extracellular peptide on the artificial membrane lacks a compelling demonstration. Initially, constructs featuring interesting peptides like somatostatin and GLP1 are developed and their subsequent utilization to modify cell behavior would significantly broaden the scope and enhance the overall impact this new technology.

Response: While we agree that the use of this technique to modify cell behavior or other biological applications is very interesting, our current focus lies in the application of the technology to artificial cells and therefore in vitro applications. Experiments regarding the utilization of this technique for creating interactions with living cells are currently underway in our lab. We would like to point out that, in order to show interaction with cells and biological activity, we will have to prove that the peptide retains its biological activity as an insert in the hemolysin loop and show that it can bind to cells, which would likely involve screening several different linkers and constructs. A cell line that is reactive to the translocated peptide will have to be identified and an assay for receptor activation will need to be developed and optimized. As such, we believe that the comprehensive characterization and evaluation that would be required for demonstrating and unequivocally validating modified cell behavior due to hemolysin translocation of a biologically active peptide is beyond the scope of our current work and will be addressed in a future study.

Comment 7:

From the data in Figure 4c-d, the authors come to the conclusion that the longer and more flexible peptide (Figure 4d) increases the frequency of pore blockage compared to the short peptide (Figure 4c). However, the curves suggest the opposite as in Figure 4c there are more low current events compared to in Figure 4d where the current is constantly high throughout the measurement.

Response: We thank the reviewer for pointing this out and have repeated the current trace measurements and revised the manuscript accordingly. The updated figure 4c-d more clearly depicts an increase in current blockage events when using the larger insert.

Comment 8:

In Figure 5 a control in the absence of the alfa-HL pores that shown background level signal from the Hyper-7 sensor is missing. In addition, the analysis method and the statistics of the quantification are not described.

Response: We thank the reviewer for their comment. A control that shows background level signal from the Hyper 7 sensor can be found in Fig. 6 C, bar 1.

The statistics of the quantification are described in the reproducibility statement:

Statistically significant differences in Fig. 6c are indicated based on an independent t-test (two-tailed): ***P < 0.001; **P < 0.01; NS, not significant. Specifically, 1 (GUVs in artificial tissue in the absence of glucose) vs 2 (GUVs outside of artificial tissues in the presence of glucose) P = 0.0830. 1 (GUVs in artificial tissue in the absence of glucose) vs 3 (GUVs in artificial tissue in the presence of glucose) P = 0.00351. 2 (GUVs outside of artificial tissues in the presence of glucose) vs 3 (GUVs in artificial tissue in the presence of glucose) P = 0.000210.

Comment 9:

The manuscript is missing a conclusion that contextualizes the results within the existing literature.

Response: We thank the reviewer for pointing this out and have revised our concluding paragraph in the manuscript to provide a more contextualized interpretation of our findings within the existing literature.

Minor Points:

Comment 10:

Fig. 1 gives the false impression that in general the alfa-HL is expressed inside the vesicles. For the initial experiments in Fig. 2-4 the figures should illustrate that alfa-HL is added as purified protein externally and that the designed peptide is presented into the lumen of the vesicle. At the same time this should be clearly stated in the text.

Response: We thank the reviewer for the comment. Firstly, figure 1 is intended to illustrate our aim conceptually. In the revised manuscript we have added additional experiments demonstrating internal expression and insertion inside vesicles. In Figure 2-4, it is true that the hemolysin is added to the outside, and the cartoon clearly indicates orientation of the pore such that it is inserting from the outside. However, we understand that there is potential for confusion, and so in the revised manuscript we have now added statements in the text and figure captions to indicate how the hemolysin was added and its orientation.

Comment 11:

The authors should clarify that the C-terminal GFP is positioned on the opposite side of the membrane to the inserted peptide.

Response: We have added a clarifying statement to indicate the position of the GFP with respect to the insert. We thank the reviewer for the suggestion.

Reviewer #2 (Remarks to the Author):

The authors use the ability of the bacterial pore-forming protein alpha-hemolysin to translocate a segment of its own polypeptide chain across lipid bilayers (as part of the pore assembly process) to move polypeptide sequences into the interior or to the surface of giant unilamellar vesicles (GUVs). This is not new, but well worth further exploration. In addition, they explore the ability of the translocated chains to connect and form network-like structures from GUVs, which is new and an intriguing means to form tissue-like materials. However, the paper needs revision to bolster the data quality and scientific presentation, and thereby substantiate the conclusions.

Comment 1:

The Methods section must be improved to ensure reproducibility and the validity of conclusions. Information is missing, for example, on the sequences of the protein constructs that have been used, and whether monomers or heptamers of alpha-hemolysin were used in the various experiments.

Response: We thank the reviewer for their comments. Sequences of the peptide inserts can be found in the SI. In addition, we have now added the full amino acid sequence of α HL with the position of the insert highlighted. Plasmid backbones and expression vectors can also be found in the SI. In all experiments, monomers of α HL were used and a clarifying statement has been added to the methods section to make this clear.

Comment 2:

Evidence should be provided for the presence of a disulfide bond in the somatostatin-containing mutant.

Response: We thank the reviewer for pointing this out and have added evidence for the presence of a disulfide bond to the SI along with a statement in the main text. (Supplementary Fig. 2) In order to show the presence of a disulfide bond in the somatostatin-containing mutant, purified protein was tested for the presence of free thiols with Ellman's reagent following a procedure from the literature (*Anal Bioanal Chem* 373, 266–276 (2002)). The data shows that α HL with the somatostatin insert does not react with Ellman's reagent indicating that the two cysteines present in the sequence have formed a disulfide bond. Treating the same sample of protein with TCEP makes it reactive to Ellman's reagent after removal of TCEP.

Comment 3:

Quantitative analysis of the fluorescence changes over time during both the dye leakage and the antibody-nanopore co-localisation experiments would be informative.

Response: We thank the reviewer for this comment and completely agree. As per the reviewer's suggestion we quantified the leakage assay data for each hemolysin mutant using image analysis. Cy5 encapsulating GUVs were treated with each of the engineered hemolysins described in the manuscript and fluorescence was measured over time. Every 45 minutes, a population of GUVs ($n > 10$) was imaged and analyzed for Cy5 fluorescence. Mean fluorescence intensity at different timepoints was calculated

for each hemolysin mutant and for untreated control GUVs. (Supplementary Fig. 3) Interestingly we could show that insert length affects leakage rate. For instance, with the 6XHis-tag inserts, increased linker length leads to slower leakage of Cy5. At the same time, insert size is not the only factor influencing leakage. We could observe for instance that charge plays an important role. aHL with the K3 insert induces leakage much faster than aHL with the E3 insert. We hypothesize that this might be due to electrostatic interactions of the negatively charged loop of the E3 pore with the negatively charged Cy5 dye, slowing down leakage.

We performed a quantitative analysis of the antibody assay as per the reviewer's suggestion (Supplementary Fig. 6). For each protein described in this work, GUVs encapsulating an antibody directed against the peptide insert were generated. The GUVs were treated with the respective aHL protein, and the resulting images analyzed in ImageJ. To quantify the increase of membrane fluorescence, a fluorescence plot profile across the whole GUV was created. A ratio of membrane fluorescence to lumen fluorescence was calculated and plotted in Supplementary Fig. 6.

Comment 4:

The current traces in Fig. 4 are not convincing with substantial fluctuations in the baselines. Only one trace has been shown for each mutant, i.e. it is not clear whether experiments were repeated. Statistically significant mean conductance values for each mutant should be reported.

Response: We are grateful to the reviewer for pointing out these issues, particularly because this is a new technique for our laboratory. We have repeated the current trace experiments shown in the original manuscript. For Fig. 4 d we were able to measure current traces with much less baseline fluctuation and again would like to thank the reviewer for pointing this out. Current trace measurements for aHL with the L-6XHis-L and L₂-GLP1-L₂ insert were repeated 3 times for each protein. The resulting current traces were used to calculate a mean conductance value as suggested by the reviewer.

Comment 5:

Respect should be paid to prior literature. For example, the translocation of long segments of polypeptide chain by the same mechanism as that exploited by the authors and access of the chain to an enzyme in the recipient compartment was demonstrated a decade ago (L. Harrington et al. PNAS, 110, E4417-26 (2013). L. Harrington et al., ACIE, 54, 8154–8159 (2015)).

Response: We sincerely apologize for not citing these two works and thank the reviewer for bringing them to our attention. We have now cited these two publications and updated the manuscript to place our work in better context with respect to the prior literature. We would like to note that, to our understanding, in Harrington et al.'s work a single peptide gets translocated during formation of a heteromeric pore in which only one of the 7 hemolysin monomer has the peptide insert. The difference to our work would be that we investigated peptide translocation with homomeric pores, where each pore, upon formation, translocates 7 copies of the same peptide, representing a much increased translocation efficiency per pore.

Comment 6:

The aggregation of vesicles to form tissue-like materials has also been investigated previously. One approach uses an electrostatic mechanism related to that explored by the authors; 'colonies' of GUVs

with negative surface charge were produced by aggregation with poly-L-arginine (P. Carrara et al. *Chembiochem* 13, 1497–1502 (2012), as cited by the authors). A more sophisticated approach is DNA-directed assembly (M. Hadorn et al. *Langmuir* 32, 3561–3566 (2016)).

Response: We thank the reviewer for mentioning this work and have now added a reference to the Langmuir paper to our manuscript.

Comment 7:

Finally, the authors might comment on whether they think molecules can diffuse from one GUV to another through two interacting nanopores containing loops of opposite charge, an idea which also has been explored previously (S. Mantri, et al. *Nature Commun* 4, 1725 (2013)).

Response: We thank the reviewer for bringing up this interesting point and have added a comment on this to the manuscript as well as a reference to S. Mantri et al.'s work. While we do think that there might be some diffusion from one GUV to another through interacting nanopores, more evidence would have to be collected supporting this. In contrast to S. Mantri et al.'s work, where adjacent nanopores are covalently linked, we assume that in our approach there is likely more flexibility at the mouth of interacting nanopores, seeing as they are only connected through reversible electrostatic interactions.

Reviewer 1:

Comment 1:

As also pointed out the by Reviewer 2, the formation of prototissues based on electrostatic interactions is not a convincing application of the system

Response:

We have read the reviews from the last round and are not sure how reviewer 1 came to this conclusion. Regarding the prototissue experiment, Reviewer 2 stated that “In addition, they explore the ability of the translocated chains to connect and form network-like structures from GUVs, which is new and an intriguing means to form tissue-like materials.”

Comment 2:

The formation of prototissues based on electrostatic interactions is not a convincing application of the system. The same thing has already been achieved with lipids of opposite charge and the current system is a more complicated way of doing this.

Response:

While we appreciate the comment, we respectfully disagree with reviewer 1. While several methods to achieve formation of prototissues have been reported in the literature (many of which we cited in this manuscript) our system is genetically encodable which has not been reported before. From an artificial cell perspective, self-encoded assembly of vesicles into prototissues offers several advantages over simply using vesicles that are composed of lipids of opposite charge. We have tried to make this point clear in the manuscript. For instance in the conclusion we state: “Current methods typically achieve artificial cell interactions by chemically modifying the outer membranes of giant vesicles.^{45,46} This contrasts with our study, where we have genetically encoded membrane functionalization. The ability of individual artificial cells to encode their own extracellular membrane modification opens new ways for programming interactions between artificial cells and their environment.”

Comment 3:

The value of modifying the GUVs with peptides will come from using bioactive peptides that can engage specific cell receptors. The authors argue that the application with cellularly active peptides would require more optimization but this is an important point if this system is going to be used by others. In the current form the work lacks in novelty to be published in Nature Communications and after addressing the more technical concerns the manuscript could be published in a more specialized journal such as Communications Biology.

Response:

We respectfully note that this comment was addressed in the previous round of revision as follows:

Comment 6:

The application of a bioactive extracellular peptide on the artificial membrane lacks a compelling demonstration. Initially, constructs featuring interesting peptides like somatostatin

and GLP1 are developed and their subsequent utilization to modify cell behavior would significantly broaden the scope and enhance the overall impact this new technology.

Response: While we agree that the use of this technique to modify cell behavior or other biological applications is very interesting, our current focus lies in the application of the technology to artificial cells and therefore in vitro applications. Experiments regarding the utilization of this technique for creating interactions with living cells are currently underway in our lab. However, we believe that the comprehensive characterization and evaluation that would be required for demonstrating and unequivocally validating modified cell behavior due to hemolysin translocation of a biologically active peptide is beyond the scope of our current work and will be addressed in a future study. In order to show interaction with cells, we would have to prove that the peptide retains its biological activity as an insert in the hemolysin loop, show that it can bind to cells which would likely involve screening several different linkers and constructs. A cell line that is reactive to the translocated peptide would have to be created and an assay for receptor activation would need to be developed.

Comment 4:

The authors point out that permeabilization is a desired feature but just as well may become problem smaller components dilute into the environment.

Response:

We respectfully note that the reviewer already made this point in the first round of revision. We responded and correspondingly modified the manuscript. From the first round of revision:

Comment:

Another weakness of the approach is that the presentation of the peptide is always coupled to pore formation, which results in membrane permeabilization.

Response: We thank the reviewer for pointing this out and have added a sentence to the manuscript making it clear that our approach results in membrane permeabilization. That being said, we wish to point out that membrane permeabilization is often desired in the construction of artificial cells to enable the transport of small molecules into and out of the vesicle. Such transport has been used in past work to improve internal protein translation (see *Proc. Natl. Acad. Sci. USA* 101, 51, (2004): 17669-17674). Furthermore, the permeabilization we observe appears to be controllable by the nature of the linker sequence, and future work may be able to use the inserts to gate membrane permeability. We have added a few sentences to the manuscript discussing these points in more detail.

Comment 5:

The data in Supplementary Fig. 3 should be moved to the main text as it provides important information on the performance of the system. The statistical significance of this data needs to be verified. The error bars in this figure are very large and the GUVs seem to be leaking even in the absence of alfa-HL. The manuscript has numerous problems concerning the labeling of the figures

Response:

While we agree that the data in Supplementary Fig. 3 does provide more insight on the system, we do not think that it is necessary to move it into the main text. Inducing leakage is not the goal of this system and as such we think quantification of GUV leakage is better placed in the SI.

The error bars in this figure are high because the formation of GUVs naturally results in GUVs of different sizes with varying encapsulation efficiency. We do not understand and are confused by the reviewer's statement that GUVs seem to be leaking even in the absence of aHL. The graph shows that Cy5 fluorescence intensity remains constant in the absence of aHL. With respect to labeling of the figures, we have done a thorough check of the figure labels and have adjusted the manuscript accordingly.

Comment 6:

The figure legends don't match with the results and don't provide sufficient details to understand the results. For example, Supplementary Fig. 1 does not show the leakage of Cy5 from the GUVs over time and the time point at which the data is reported is not mentioned. A full timeline, with the initial GUV and the change over time must be provided. All figures should be reviewed with care.

Response:

Supplementary Fig. 1 does not include leakage over time because this experiment was quantified in Supplementary Fig. 3. It should be pointed out that Supplementary Fig. 3 is the figure the reviewer mentions in the previous comment. As a reminder, in Supplementary Fig. 3, we show data on vesicle leakage over time for all the different aHL proteins as per the reviewer's request from the previous round of revision.

The reviewer states "the time point at which the data is reported is not mentioned". However, in Supplementary Fig. 1, we state "Shown are representative GUVs after **90 min of incubation** with GFP- α HL fusion protein at RT".

Comment 7:

The quantification of the data in Figure 3 is also still missing. Currently, the conclusions still rely on single images.

Response:

The quantification of this data was added during the previous round of revision in response to reviewer comments. It was added in the revised version as Supplementary Fig. 6. We also changed the text in the main manuscript to show that we have quantified these results. And have referenced the relevant supplementary figure. To make this even clearer to the reader, we have now also added a reference to supplementary figure 6 to the figure subtitle of figure 3.

Comment 8:

I appreciate the demonstration of the in vitro protein synthesis in a second set of experiments. This data should be moved to the main text.

Response:

We believe the reviewer is referring to Supplementary Fig. 5. We do think that showing that a GLP1 antibody can bind GLP1 peptide expressed and translocated through a GUV membrane by aHL is further proof that our system can be used to translocate peptides from inside artificial cells to the outside and vice versa. However as this data further confirms what has already been shown in the manuscript we believe that this figure should remain in the SI.

Comment 9:

Figure 6 needs to be revised. The graph is missing its y-axis label. The data analysis and statistical analysis are not clear. The GUVs that are claimed to be in the white square are not visible in any image. I would recommend using a membrane dye to clearly visualize the GUVs. The samples that are labeled as 1/2/3 in the graph are not described in the text.

Response:

We thank the reviewer for pointing out these issues. The Y-axis label of this graph was described in the title of the figure, but we agree that the axis should have been labeled on the graph itself. We have now added the Y-axis label “Fluorescence Intensity”.

We also added a figure to the SI with increased brightness in the green channel to make the GUVs in white squares more visible. The samples labeled 1/2/3 are described in detail in the figure subtitle. We have edited the text to make this clearer by adding an additional reference to this figure in the main text.

The reviewer had asked about the statistical analysis in the first round of revision. We point out that the statistical analysis is described in the statistics and reproducibility statement of the main manuscript. However, we have now added an additional explanation concerning the statistical analysis to all the relevant figures.

Comment 10:

The material and methods are not sufficiently detailed for others to reproduce the data. The image analysis routine for the permeability assay and the communication is not clear.

Response:

The methodology for the permeability assay was described in Supplementary Fig. 3. We have reviewed the materials and methods section and the image analysis routine for the permeability assay and we respectfully feel that the methods have been clearly explained such that others can repeat the experiments.

Reviewer 2:

Reviewer #2 (Remarks to the Author):

The authors have made an unusually strong and largely successful attempt to reply to the points raised by both Reviewers. The outcome is a high-quality piece of work with interesting implications in synthetic biology, drug delivery into cells, and more.

This Reviewer has a few remaining comments, that could be left to the authors to consider:

Response:

We thank the reviewer for this assessment and their review of this manuscript.

Comment 1:

Abstract and title inflation should be controlled at the editorial level; they are all some readers look at. The title here is fine. However, the Abstract does not make it clear that the basis of the work is an elaboration of a previous finding: that aHL can translocate fused sequences across bilayers. The reported work is certainly very interesting indeed, but it is incorrect to imply that a new property of aHL itself has been discovered.

Second, the authors end the Abstract by stating that they have made "artificial tissue like structures capable of signal transduction". This last phrase implies, subtly, that the engineered pores are involved in signaling, which they are not. Presumably any means of clustering the vesicles would work because the H₂O₂ likely passes through the external solution and enters the receiver "cells" before dilution as indicated in Figure 6.

Response:

We thank the reviewer for pointing this out and have made changes to both the abstract and the manuscript in response to the reviewer's comments.

The abstract was changed to not imply that a new property of aHL has been discovered. We removed statements such as "can be engineered" and "this method enables" which imply this is somehow a new concept in aHL design, and instead have simply reported what we have done in the work.

In addition, we have changed the final statement of the abstract to remove any implication that engineered pores could be involved in signaling. The new abstract reads as follows:

The development of artificial cells has led to fundamental insights into the functional processes of living cells while simultaneously paving the way for transformative applications in biotechnology and medicine.^{1,2,3,4} A common method of generating artificial cells is to encapsulate protein expression systems within lipid vesicles.⁵ However, to communicate with the external environment, protein translocation across lipid membranes must take place. In living cells, protein transport across membranes is achieved with the aid of complex translocase systems which are difficult to reconstitute into artificial cells.⁶ Thus, there is need for simple mechanisms by which proteins can be encoded and expressed inside synthetic compartments

yet still be externally displayed. Here we present a genetically encodable membrane functionalization system based on mutants of pore-forming proteins. We modified the membrane translocating loop of α -hemolysin to translocate functional peptides up to 52 amino acids across lipid membranes. Full membrane translocation occurs in the absence of any translocase machinery and the translocated peptides are recognized by specific peptide-binding ligands on the opposing membrane side. Engineered hemolysins can be used for genetically programming artificial cells to display interacting peptide pairs, enabling their assembly into artificial tissue-like structures.

Comment 2:

The differences in dye leakage from GUVs caused by the various mutants is not considered in detail. We note first that monomers were used. Considerations include:

(i) the binding rate and affinity of aHL to the GUVs. This is especially important because an active heptamer must be formed; (ii) the lag period, which can be many minutes depending on the conditions and arises from (a) prepore assembly (b) insertion of the prepore. In the paper, the most obvious lag is in Figure S3c; (iii) the rate of transport once a membrane-spanning heptamer has formed. This depends on the nature of the transported molecule and cannot simply be estimated from the unitary ionic conductance (e.g. Wu, Y., et al. Permeation of styryl dyes through nanometer-scale pores in membranes. *Biochemistry* 50, 7493-7502 (2011)).

Response:

We thank the reviewer for bringing this to our attention and have added a paragraph to the relevant figure to highlight these important points. The assay we used to analyze leakage over time through the different aHL pores described in the manuscript is not a quantitative way to measure rate of transport across aHL pores. It merely serves as a useful comparison of the different mutants reported in this work and, as the reviewer rightfully points out, is dependent on binding rate, prepore assembly and speed of insertion of the respective pore into the membrane.

We added the following paragraph to Supplementary Fig. 3:

While the leakage data shown in this figure can be used as a baseline for the design of new inserts, the data does not represent a quantitative measurement of small molecule transport across the respective pores. The leakage data presented is influenced by various additional variables such as the binding affinity of the monomer to the GUV membrane, as well as subsequent prepore assembly and prepore insertion.